# Modeling Artificial Light Exposure after Vegetation Trimming at a Marine Turtle Nesting Beach

**Mark A. Barrett** * and **Kristen Nelson Sella**

Florida Fish and Wildlife Conservation Commission, Fish and Wildlife Research Institute, Information Science & Management, Center for Spatial Analysis, 620 South Meridian Street, Tallahassee, FL 32399, USA; kristen.nelsonsella@myfwc.com
* Correspondence: mark.barrett@myfwc.com

**Abstract:** Light pollution caused by poorly directed artificial lighting has increased globally in recent years. Artificial lights visible along marine turtle nesting beaches can disrupt natural brightness cues used by hatchling turtles to orient correctly to the ocean for their offshore migrations. Natural barriers, such as tall dunes and dense vegetation, that block coastal and inland lights from the beach may reduce this disruption. However, coastal areas are often managed toward human values, including the trimming of vegetation to improve ocean views. We used viewshed models to determine how reducing the dune vegetation height (specifically that of seagrape, *Cocoloba uvifera*) might increase the amount of artificial light from upland buildings that reaches a marine turtle nesting beach in Southeast Florida. We incorporated three data sets (LiDAR data, turtle nest locations, and field surveys of artificial lights) into a geographic information system to create viewsheds of lighting from buildings across 21 vegetation profiles. In 2018, when most seagrape patches had been trimmed to <1.1 m tall, female loggerhead turtles nested in areas with potential for high light exposure based on a cumulative viewshed model. Viewshed models using random (iterative simulations) and nonrandom selections of buildings revealed that untrimmed seagrape heights (mean = 3.1 m) and especially taller vegetation profiles effectively reduced potential lighting exposure from three building heights (upper story, midstory, and ground level). Even the tallest modeled vegetation, however, would fail to block lights from the upper stories of some tall buildings. Results from this study can support management decisions regarding the trimming of beach dune vegetation, any associated changes in the visibility of artificial lighting from the nesting areas, and modifications to existing lighting needed to mitigate light exposure.

**Keywords:** viewshed; marine turtles; artificial light; light pollution; vegetation trimming; LiDAR

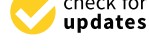



## 1. Introduction

Light pollution created by errant artificial lighting has increased rapidly and globally in recent years [1,2]. Artificial light from man-made environments can alter natural light regimes in adjacent wildlife habitats and thus affect species that live there, i.e., cause ecological light pollution [3]. For example, artificial light can modify natural behaviors (e.g., navigation, foraging) in nocturnal species [3–5]. Artificial lighting has encroached upon many natural areas, but its concentration is most evident along heavily developed coastlines, where it can impact marine turtles [6–14].

Light pollution on marine turtle nesting beaches can disrupt natural cues used by hatchling turtles to orient to the ocean for their offshore migration [15]. Hatchlings typically move away from the tall, dark silhouette of dunes and vegetation and move toward the lighter horizon of the ocean [16,17]. However, artificial lights visible from a nesting beach can produce a light-trapping effect and disorient hatchlings, causing them to travel landward or in a circuitous manner; disorientation forces hatchlings to expend energy that is crucial for their offshore migration and increases their risk of predation and

dehydration [18,19]. This harmful impact on marine turtles during their early life history can decrease population recruitment at nesting beaches near developed areas [12,20,21].

Efforts to mitigate light pollution near turtle nesting beaches include modifying lights by altering their wavelength with suitable bulbs, by directing their beam downward, or by reducing their radiance with shielding [22,23]. However, even with such modifications, disorientation can still be triggered by inland light sources such as skyglow—the diffused illumination of the night sky above densely developed areas [1,18,22,23]. Natural barriers, such as tall dunes and dense vegetation, can buffer or block coastal and inland lights and may even provide a silhouette preferred by nesting female turtles [6,24]. However, coastal areas are often developed for their recreational value to humans [25], which sometimes leads to management decisions that benefit the owners of the property adjacent to nesting beaches. Strategies for managing coastal vegetation can vary depending on the interests of beachfront property owners. Some property managers of beachfront areas want to rebuild natural dunes and associated vegetation to protect against storm damage and erosion [26,27]. However, others request permits to cut or trim dune vegetation to provide unobstructed views of the shoreline and ocean to increase the human value (intrinsic and monetary) of coastal property [28,29]. Wildlife managers have expressed concern that trimming or cutting dune vegetation will increase artificial lighting impacts on marine turtles and other coastal species [6,14,30]. Yet, data are lacking that quantify the impacts of reducing vegetation profiles on the visibility of artificial light on beaches.

Conducting field experiments to measure lighting impacts before and after trimming of vegetation can prove difficult because a schedule of trimming operations is seldom available, and other variables (e.g., cloud cover, moon phase) can confound lighting measurements. Alternatively, scientists have used remote sensing tools to measure artificial lighting exposure to species and their habitats [7,10]. A viewshed tool, typically employed in a geographic information system (GIS), is a technique that uses remotely sensed data to determine the line-of-sight visibility from a particular vantage point (or observer location) to localities (or surface grid cells) across a landscape [31]. Viewsheds from multiple observers can also be combined to develop a cumulative viewshed, producing a relative frequency of visibility per locality in a landscape. A viewshed incorporates a landscape defined by topographical features built with remotely sensed information, such as high-resolution LiDAR data that can be used to create three-dimensional images of the earth's surface, including elevation values [32]. These created surfaces are generally referred to as digital elevation models (DEM). The DEMs used in viewsheds can be solely based on ground elevation (e.g., digital terrain model [DTM]), but more realistic analyses include above-ground physical features (both artificial and natural), such as buildings and vegetation (e.g., digital surface model [DSM]). Viewsheds have been used in several animal ecology studies to model relationships between animals and their environment [31], including various levels of observation height and vegetation patterns [33] and predictions of light exposure to nesting wildlife [9,34].

The aim of our study was to model how modifications to the height of beach dune vegetation (specifically that of seagrape, *Cocoloba uvifera*) affect the amount of urban artificial lighting that is visible at a marine turtle nesting beach in Southeast Florida. To accomplish this, we used LiDAR data to build DEMs that represented a landscape comprised of buildings that were sources of potential artificial light and various dune vegetation profiles. We incorporated these data into a cumulative viewshed model to predict the impacts of lighting exposure from buildings on turtle nesting sites. We ran iterative simulations of viewsheds from randomly selected buildings to model the mean percentage of turtle nests potentially exposed to different levels of beachfront lighting based on 21 dune vegetation profiles. We also nonrandomly selected buildings and ran models of cumulative viewsheds to exemplify extremes in differences in nesting beach exposure to lighting among vegetation profiles. The results of this study can be used to inform management decisions regarding the extensive trimming of beach dune vegetation and any associated shifts in the visibility of artificial light.

## 2. Materials and Methods

### 2.1. Study Site

The study site is delineated on a landscape between coastal range monuments R-176 and R-183 [35] in Delray Beach, Florida, USA (Figure 1). The site is 2.5 km² in area, in which beach and dune cover are each 0.1 km², and the urban cover is 2.2 km² in area. The beach is open to the public and has been documented as a marine turtle nesting area since 1984 [36]. The urban area contains residential (including rental and vacation properties) and commercial buildings. In Delray Beach, the mean annual precipitation is 154.9 cm, the mean annual temperature is 23.9 °C, and the mean elevation is 2.7 m.

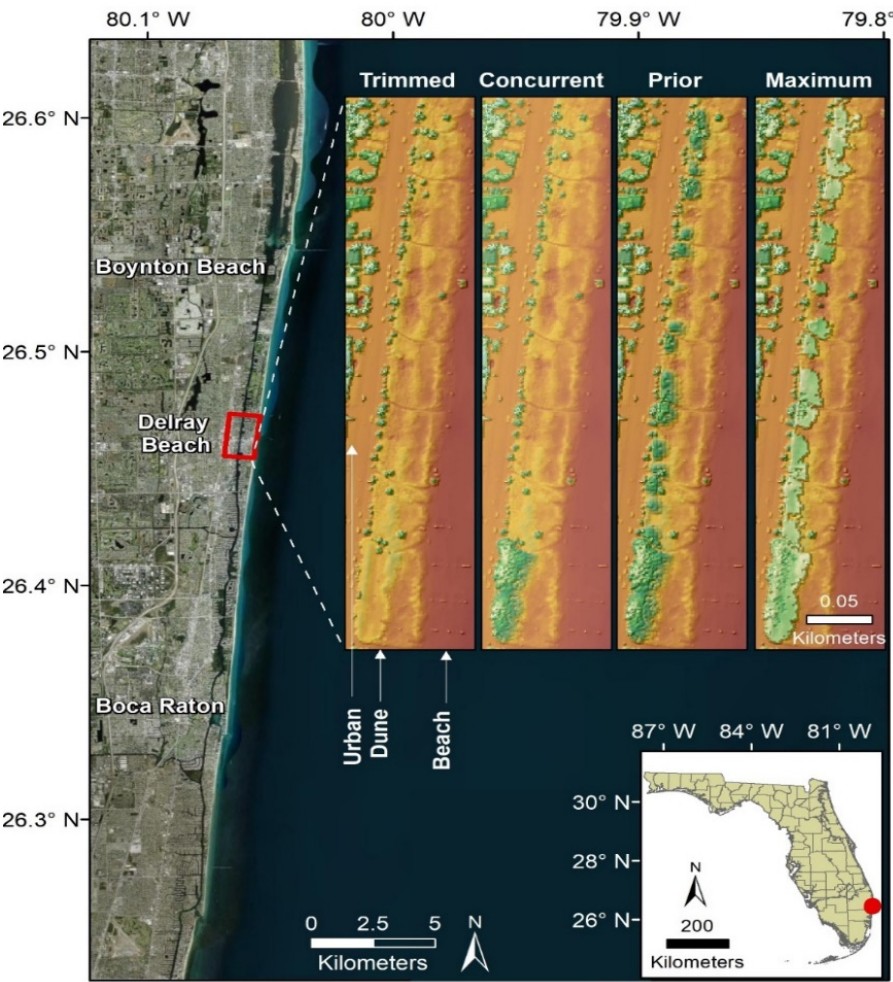

**Figure 1.** Map of the study area (red polygon) that includes beach, coastal dune, and urban land cover classes located in Delray Beach, Florida, USA (inset). Examples of height profiles of dune vegetation from the southern end of the study area were created from LiDAR data (1 m resolution); height profiles include trimmed to ~1 m tall, concurrent with marine turtle nest data in 2018, prior based on 2004 LiDAR data, and a randomly generated maximum height between 6 and 8 m. Satellite imagery is from 2020–2021 World Imagery (Esri base map).

The dune vegetation comprises seagrape, buttonwood (*Conocarpus erectus*), sable palm (*Sabal palmetto*), saw palmetto (*Serenoa repens*), cocoplum (*Chrysobalanus icaco*), bay cedar (*Suriana maritima*), beach naupaka (*Scaevola taccada*), sea oats (*Uniola paniculata*), dune sunflower (*Helianthus debilis*), dune panic grass (*Panicum amarum*), and saltmeadow cordgrass (*Spartina patens*). Vegetation heights across all species ranged approximately from <1 m to 13 m. Seagrape is a broad-leaved plant that typically forms a multistemmed vase shape and can grow to 15 m tall [37]. In the study area, seagrape is trimmed according

to specifications in the dune vegetation trimming plan of the City of Delray Beach [38]. The plan identified 147 patches of seagrape in the study area and described seagrape heights before trimming using general ranges (e.g., 1–3 ft, 4–6 ft) with minimum and maximum heights at 1 ft (0.3 m) and 20 ft (6.1 m), respectively. By 2018, all but three patches of seagrape had been trimmed to either 0.9 m (3 ft) or 1.1 m (3.5 ft) per the specifications of the vegetation trimming plan to increase the viewscape for beachfront properties [38].

### 2.2. Data Collection

The Florida Fish and Wildlife Conservation Commission (FWC) authorized the collection of turtle nesting data by marine turtle permit holders for this study. Data were collected in 2018 between 1 May and 13 August [39]. Seventy-four loggerhead turtle (*Caretta caretta*) nests were observed in the study area. Permit holders collected geographic coordinates for nest sites using handheld GPS receivers and measured the distance (m) to the edge of the nearest dune. To assess possible spatial errors in nest locations, we compared estimates of distance from each nest to the nearest dune from the field observations to those computed in the GIS using Euclidean distance measures. We removed those distance estimates (*n*= 6) that differed by >20 m. As a result, we retained 68 nests, which had a median difference of 2.9 m between the two distance estimates. We therefore buffered the nests by 3 m in producing polygons for use in analyses. We also examined the disorientation of turtle hatchlings, determined during field surveys by the marine turtle permit holders, who visually examined tracks of hatchlings in the early morning after a nest emergence [40], thus assuming all hatching occurred at night (i.e., absence of daylight). When a hatchling's tracks did not travel toward the ocean, the hatchling was considered disoriented. We computed the disorientation rate as the percentage of nests with emergent hatchlings (*n* = 56) with observed disorientation.

We lacked actual artificial lighting data for our study area during 2018. Therefore, to estimate potential sources of artificial lighting, we used information from two artificial lighting surveys conducted by Ecological Associates Inc. (Jensen Beach, Florida, USA) during marine turtle nesting season in 2020 on 27 March and 27 April [40]. For the surveys, an observer walked along the beach (between 2300 and 0000 h) taking photographs and recording geographic coordinates at points at which artificial lights (those not modified to minimize light trespass to the beach) were visible from buildings or, less often, from other features (e.g., streetlights, decorative string lights). Within the study area, lights were visible at different building heights (ground level to the upper story) from 20 buildings in the first survey and 5 buildings in the second survey. We incorporated this information into iteratively simulated models that used different numbers of buildings representing potential light sources in viewshed analyses (described below).

To construct DEMs, we obtained LiDAR data from the National Oceanic and Atmospheric Administration (NOAA). To develop DEMs concurrent with turtle nesting surveys, we used LiDAR data collected in 2018 by NOAA's National Geodetic Survey Remote Sensing Division (NOAA, Silver Spring, MD, USA). To account for a specific vegetation profile (see below), we used LiDAR data collected in 2004 by the Joint Airborne LiDAR Bathymetry Technical Center of Expertise (NOAA Coastal Management, Charleston, SC, USA). Both data sets were in LAS file format, had a vertical accuracy of ±0.15 m, had a horizontal accuracy of ±1.0 m, and were set to UTM Zone 17 NAD 83 horizontal datum and NAVD88 vertical datum. In preparation for building surface models in a GIS, we converted the LAS files to multipoint data for first, last, and ground LiDAR returns using the 3D Analyst extension in ArcMap 10.5.1 (Esri, Redlands, CA, USA).

### 2.3. Surface Models, Buildings, and Vegetation Profiles

From the multipoint data, we created 1 m (pixel size) resolution DEMs using the Natural Neighbor tool in the Spatial Analyst ArcMap extension. Final DEMs were 32-bit floating-point TIF files that included a DTM based on the LiDAR ground return classification and a DSM based on LiDAR first returns. The DEMs based on 2018 data were used as

the base layers for analyses and included urban, dune, and beach areas; the DEMs based on 2004 data were used only to estimate and create a prior-to-trimming dune vegetation profile. To minimize false line-of-sight obstructions, we flattened water bodies to a 0 m elevation, and we manually removed power lines (those visible in the DSM) by creating 3 m buffers around them and replacing their DSM values with corresponding DTM values. To obtain the heights of buildings and vegetation, we subtracted the DTM from the DSM. We also created a surface model based on LiDAR last returns as an intermediary step in extracting building footprints.

We extracted buildings from the DSM to represent possible artificial light sources. First, we subtracted the surface built from LiDAR last returns from the DSM; differences in pixel values for solid surfaces (e.g., ground and rooftops) should be near 0 m, whereas differences for more penetrable surfaces (e.g., tall vegetation), in general, were expected to be greater. Therefore, we retained any pixels with differences of <0.5 m to account for slight angular inaccuracies of LiDAR around building edges. From these pixels, we retained only those with heights >3 m (i.e., DSM—DTM), the minimum height of a single-story building in our study area. We then reclassified the pixels to a value of 1, regionally grouped them using an 8-neighbors parameter, and retained groups having a count of >50 pixels, roughly the footprint size of a smaller building. We then converted the pixel groupings (i.e., building footprints) to polygons, removed any donut holes with the Aggregate tool, and converted the polygon edges to points using 10 m spacing, resulting in 12,858 points. The points represented building edges on which light sources might exist and were considered the observer input in our viewshed models.

To determine the effects of vegetation profiles on the visibility of buildings from the beach, we created various profiles of seagrape height. Of the 147 patches of seagrape identified in the Delray Beach vegetation plan [37], >80 were very small patches or single plants not easily discernible in the GIS. Therefore, across patch areas that were identifiable in the GIS, we developed 21 profiles of seagrape height based on the 2018 DSM (concurrent with turtle nesting data), the 2004 DSM (prior-to-trimming heights), the proposed trimmed height (0.9 m or 1.1 m), 17 height profiles at 0.3 m (1 ft) intervals from 1.2 m to 6.1 m, and a maximum height, produced by randomly assigning heights between 6.1 and 8.0 m across pixels containing seagrape because some seagrape heights reached 8 m in 2004 LiDAR data (Table 1, Figure 1). We did not manipulate vegetation occurring outside of seagrape patches.

**Table 1.** Seagrape height classes ($n$ = 21) derived from LiDAR data collected in 2018 (and 2004 for the prior vegetation profile) in coastal dunes within the study site in Delray Beach, Florida, USA. Heights in seagrape patches were set to maximum heights found in the study site, to thresholds based on 0.3 m (1 ft) intervals, to a prior-to-trimming height based on 2004 LiDAR data, to a height concurrent with 2018 marine turtle nesting data, and to a trimmed height based on a city vegetation trimming plan. Mean heights (m) and standard deviations (SD) were computed for all dune vegetation $\geq$ 0.9 m in height (minimum trimmed height) within the study site.

| Height Profile | Mean (m) | SD | Description |
|:---:|:---:|:---:|:---:|
| Maximum | 6.23 | 2.04 | Random height from 6.1 m to 8.0 m |
| 6.1 | 5.22 | 1.71 | Maximum height leveled to 6.1 m |
| 5.8 | 5.03 | 1.61 | 6.1 m − 0.3 m |
| 5.5 | 4.80 | 1.51 | 6.1 m − 0.3 m × 2 |
| 5.2 | 4.61 | 1.43 | 6.1 m − 0.3 m × 3 |
| 4.9 | 4.37 | 1.34 | 6.1 m − 0.3 m × 4 |
| 4.6 | 4.14 | 1.27 | 6.1 m − 0.3 m × 5 |
| 4.3 | 3.95 | 1.22 | 6.1 m − 0.3 m × 6 |
| 4.0 | 3.72 | 1.17 | 6.1 m − 0.3 m × 7 |
| 3.7 | 3.53 | 1.14 | 6.1 m − 0.3 m × 8 |
| 3.4 | 3.30 | 1.13 | 6.1 m − 0.3 m × 9 |

**Table 1.** *Cont.*

| Height Profile | Mean (m) | SD | Description |
|---|---|---|---|
| Prior | 3.11 | 1.90 | Height based on 2004 DSM |
| 3.1 | 3.10 | 1.13 | 6.1 m − 0.3 m × 10 |
| 2.7 | 2.87 | 1.15 | 6.1 m − 0.3 m × 11 |
| 2.4 | 2.64 | 1.19 | 6.1 m − 0.3 m × 12 |
| 2.1 | 2.45 | 1.24 | 6.1 m − 0.3 m × 13 |
| 1.8 | 2.22 | 1.31 | 6.1 m − 0.3 m × 14 |
| 1.5 | 2.02 | 1.38 | 6.1 m − 0.3 m × 15 |
| Concurrent | 1.81 | 2.06 | Height based on 2018 DSM |
| 1.2 | 1.79 | 1.47 | 6.1 m − 0.3 m × 16 |
| Trimmed | 1.53 | 1.61 | Height limited to 0.9 m or 1.1 m |

### 2.4. Viewshed Models

We ran viewshed models using the Visibility tool in the Spatial Analyst extension with the following parameters: earth curvature correction, refractivity coefficient set at the default of 0.13, and an outer radius from observer points (i.e., building lights) set at 1609 m (1 mile). To avoid unnecessarily running viewsheds for all 12,858 building points, we first ran a viewshed model from points generated on the beach to determine which buildings were visible. To accomplish this, we created contours along the beach at 1 m intervals using the Contour tool in the Spatial Analyst extension and converted the generated lines to points using 10 m spacing, resulting in 1051 observer points. We ran a viewshed model from these points and retained building points ($n$ = 652 points) that intersected DSM pixels marked as visible from contour points, resulting in 73 visible buildings. We then generated building lights from these 73 buildings at three heights: upper story, midstory, and ground level. Upper story points were the 652 points (retained above) on the building roof's edge; midstory points were manually placed at the building's base near the upper story points and set to half the building height; and ground level points were placed at the midstory point location but set to 2 m off the ground. Midstory and ground-level points were not created where obstructions (e.g., vegetation, another building) occupied the position or obviously blocked the line of sight to the beach. For shorter buildings, where midstory and ground story height was equivalent, we only used the ground story points.

### 2.4.1. Cumulative Viewshed

To determine the relationship between nest site locations and potential exposure to artificial lights, we used our concurrent vegetation profile that paralleled the turtle nest survey period and ran a cumulative viewshed model for the 73 buildings at all three building heights combined. We intersected the 68 turtle nest polygons (i.e., the 3 m buffers) with the cumulative viewshed output to compute the mean frequency of visible building lights across all nest locations. We then conducted 1000 iterations of randomly generated points ($n$ = 68 per iteration; each point was buffered by 3 m) across the beach. We intersected the random point buffers with the viewshed output, computed the mean frequency of building lights, and built a frequency distribution using all iterations to represent possible frequencies of visible building lights across the beach. If turtles avoided nesting in areas exposed to light, then the mean lighting frequency of observed nests should fall below 95% of the distribution of mean expected lighting frequencies generated from the random points.

### 2.4.2. Randomly Iterated Viewsheds

To estimate nest exposure to various lighting scenarios and vegetation profiles, we simulated a multitude of viewshed models. We used the Iterations tool in ArcMap Model Builder to run separate viewsheds for each individual building for each height class (upper story, midstory, ground level) within each of the 21 vegetation profiles, resulting in 4221 viewshed models. We intersected results from each viewshed model with turtle

nest polygons, created a data matrix of the intersected results, and binarily classified nests as being exposed (1 = yes or 0 = no) to each building's viewshed. We randomly selected buildings at each of five levels (5, 10, 20, 30, 40, 50) within the matrix to account for observed lighting levels (5–20) and for the possibility of higher levels (30–50). Using R programming language [41], we employed the dplyr package [42] to randomly sample buildings and the Matrix package [43] to summarize the number of nests visible from the randomly selected buildings. We iterated this process 1000 times per level of random building selection within each vegetation profile and used the set.seed option for reproducibility among model runs. From the simulated models, we computed the mean percent (and 95% confidence intervals) of nests potentially exposed to building lights for each building height class and number (i.e., 5–50). Furthermore, we determined the effect sizes of nest exposure to artificial lights by using the trimmed vegetation class as a reference level. For each number of buildings, we calculated a simple effect size ($\Delta$) by contrasting the percentage of exposed nests in each vegetation height profile to the percentage of exposed nests in the trimmed height profile. Along with the simple effect sizes, we used the effsize package [44] to calculate Cohen's D effect size indices [45] and their 95% confidence intervals. Cohen (1992) described effect sizes (*d*) as small (*d* = 0.2–0.5), medium (*d* = 0.5–0.8), and large (*d* > 0.8), where *d* represents a difference of 1 standard deviation between means.

### 2.4.3. Nonrandomly Selected Viewsheds

We nonrandomly selected buildings to convey possible extents of vegetation effects in blocking artificial light on the entire beach (not just at the nesting sites). Initially, we ran a viewshed model per building with all heights combined using four vegetation profiles (trimmed, concurrent, prior, and maximum) and computed the area ($m^2$) of potential light exposure on the beach. We then contrasted light exposure between trimmed vegetation and the other three vegetation profiles for each building, and then we nonrandomly selected buildings with the greatest and smallest individual differences in light exposure (*n* = 10 for each). We ran cumulative viewshed models for the 10 buildings from each selection and descriptively compared summed area and the percent area of potential light exposure on the beach across the four vegetation profiles.

## 3. Results

### *3.1. Surface Model Summaries*

Based on the 2018 DTM, the mean dune height $\pm$ SD was 3.9 $\pm$ 0.5 m. In the 2018 DSM, we identified 56 relatively large patches of seagrape, which ranged in area from 21 $m^2$ to 1786 $m^2$ (mean area = 265.9 $\pm$ 300.2 $m^2$). Seagrape heights in the 2004 DSM fell mostly within the estimated height ranges prior to trimming in the Delray Beach trimming plan [37], though some seagrape pixels in the DSM were as high as 8 m (26 ft). Mean dune vegetation heights ranged from 1.5 $\pm$ 1.6 m for the trimmed class to 6.2 $\pm$ 2.0 m for the maximum class (Table 1).

We extracted 896 buildings from the 2018 DSM approximating a building density of 407/$km^2$. Under the trimmed vegetation landscape (i.e., the reference profile), 73 of these buildings were visible from the beach and had a mean upper story height of 11.6 $\pm$ 8.6 m, ranging from 3.1 to 45.2 m and having a mean footprint of 1149.5 $\pm$ 1271.1 $m^2$, ranging from 111.6 to 8163.0 $m^2$. The mean distance of the 73 buildings to the shoreline was 204.2 $\pm$ 127.3 m, ranging from 91.2 to 1008.8 m; 76.0% were within 200 m, and 8.9% were >500 m from the shoreline. The mean number of observer points (i.e., lights) per building was 15.6 $\pm$ 16.1, ranging from 1 to 86; the total number of observer points was 1233, with 652 for the upper story, 272 for midstory, and 309 for the ground level. From the artificial lighting surveys in 2020, the mean upper story height of visible buildings was 15.5 $\pm$ 12.9 m, and the mean distance from the shoreline was 283.3 $\pm$ 198.8 m.

### 3.2. Cumulative Viewshed

Based on the cumulative viewshed using our concurrent vegetation profile, the mean number of lights visible per turtle nest was 2.8, and the mean frequency of lights visible per random point was 2.5 (95% CI = 1.7–3.3). The relationship between lighting frequencies for nests and the random distribution indicated that potentially high artificial lighting areas across the beach were occupied by turtle nests (Figure 2). The disorientation rate for hatchling turtles in our study area in 2018 was 8.9% (5 out of 56).

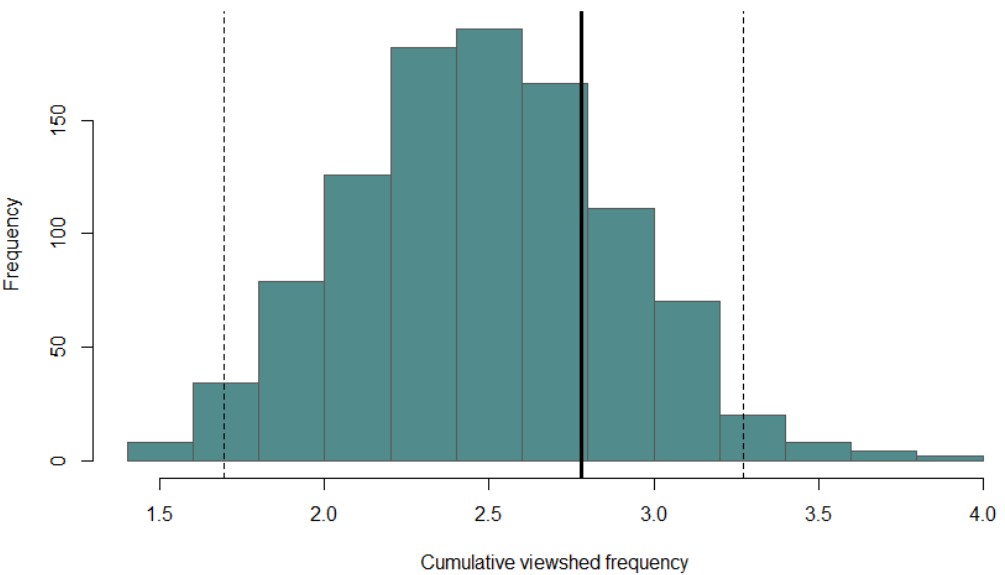

**Figure 2.** Distribution, with upper and lower 95% confidence intervals (dashed vertical lines) of the number of building lights visible across the beach based on 1000 iterations of randomly generated points (*n* = 68 per iteration) and the mean frequency of building lights (solid vertical line) visible from 68 marine turtle nest locations within the study site in Delray Beach, Florida, USA, in 2018. Potential lights were from 73 buildings at three height levels (upper story, midstory, and ground level), and visibility frequency was based on a cumulative viewshed model.

### 3.3. Randomly Iterated Viewsheds

For randomly iterated models, viewsheds from the upper stories of buildings resulted in a range of 11.5–88.9% of turtle nests being exposed to artificial lights (Figure 3), in which medium to high effect sizes (effect size Δ = 10.1–26.5%; *d* = 0.5–1.2) were evident starting around the 5.2 m vegetation height profile (Figure 4). Viewsheds from the midstory of buildings resulted in a range of 3.5–55.6% of turtle nests being exposed to artificial lights (Figure 3), in which medium to high effect sizes (effect size Δ = 9.2–29.4%; *d* = 0.5–2.2) were evident starting around the 4.9 m vegetation height profile (Figure 4). Viewsheds from the ground level of buildings resulted in a range of 0.2–4.3% of turtle nests being exposed to artificial lights (Figure 3), in which medium to high effect sizes (effect size Δ = 0.8–4.3%; *d* = 0.5–8.7) were evident starting around the prior vegetation profile (Figure 4). Notably, for all three building heights, the percent of nests exposed declined at the prior-to-trimming height having at least small effect sizes (Figures 3 and 4). Under the trimmed vegetation profile, as buildings were randomly added between 5 and 50, the percent of nests exposed increased considerably, by 61.8% and 46.8% for the upper story and midstory heights, respectively (Figure 3).

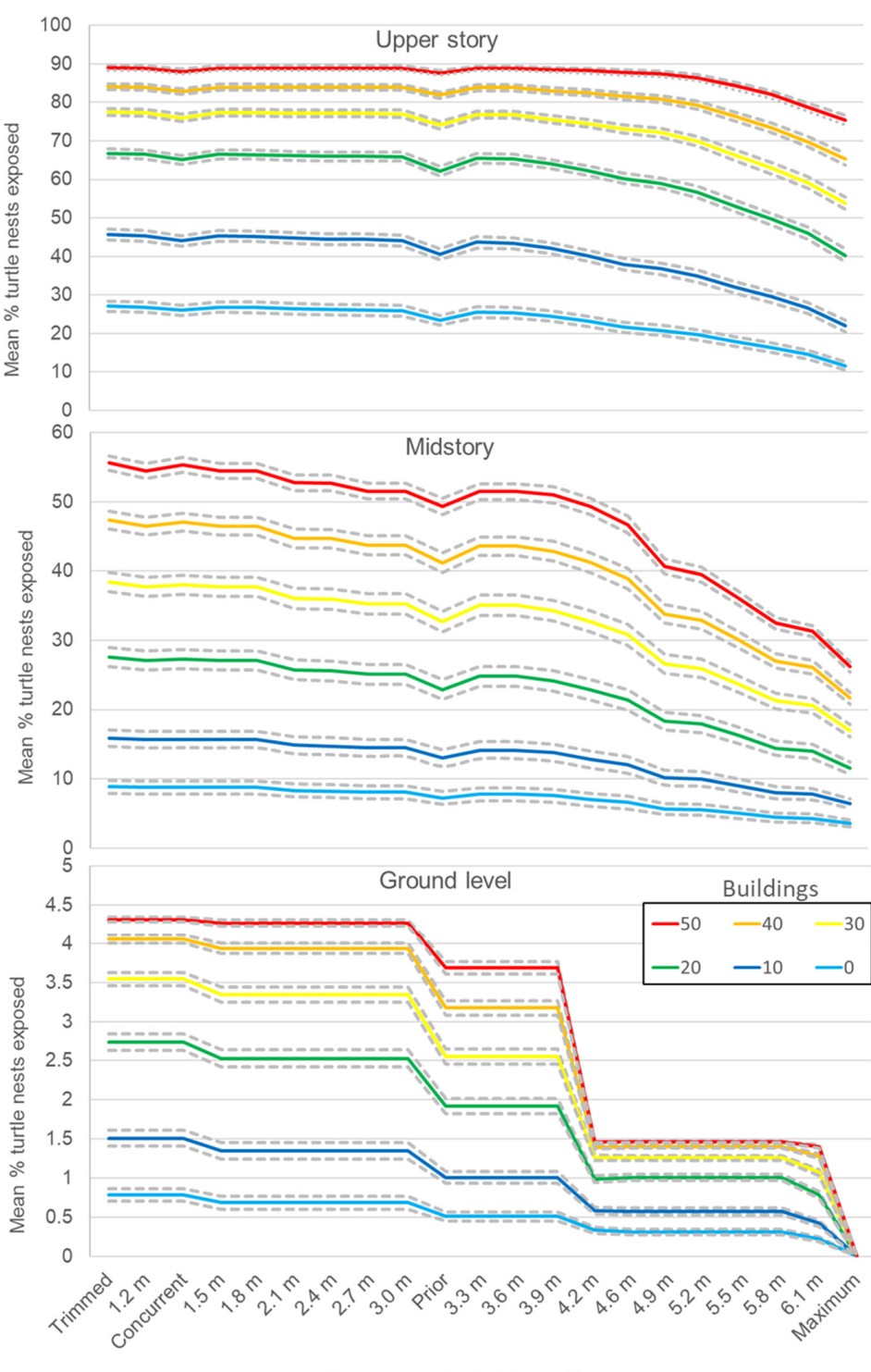

**Figure 3.** Mean percentage (solid black lines) and 95% confidence intervals (gray dashes) of loggerhead turtle nest locations (*n* = 68) exposed to artificial lights based on viewshed models from buildings for data collected in 2018 in Delray Beach, Florida, USA. Viewsheds were run at three building heights (upper story, midstory, ground level) using random subsamples of buildings (5–50; iterated 1000 times for each subsample) within multiple height profiles of dune seagrape: trimmed to ~1 m, based on concurrent 2018 LiDAR data, prior based on 2004 LiDAR data, generated maximum of 6.1–8.0 m, and generated classes in 0.3 m (1 ft) increments from 1.2 to 6.1 m. The abscissas are ordered from shortest to tallest mean vegetation height in the dunes.

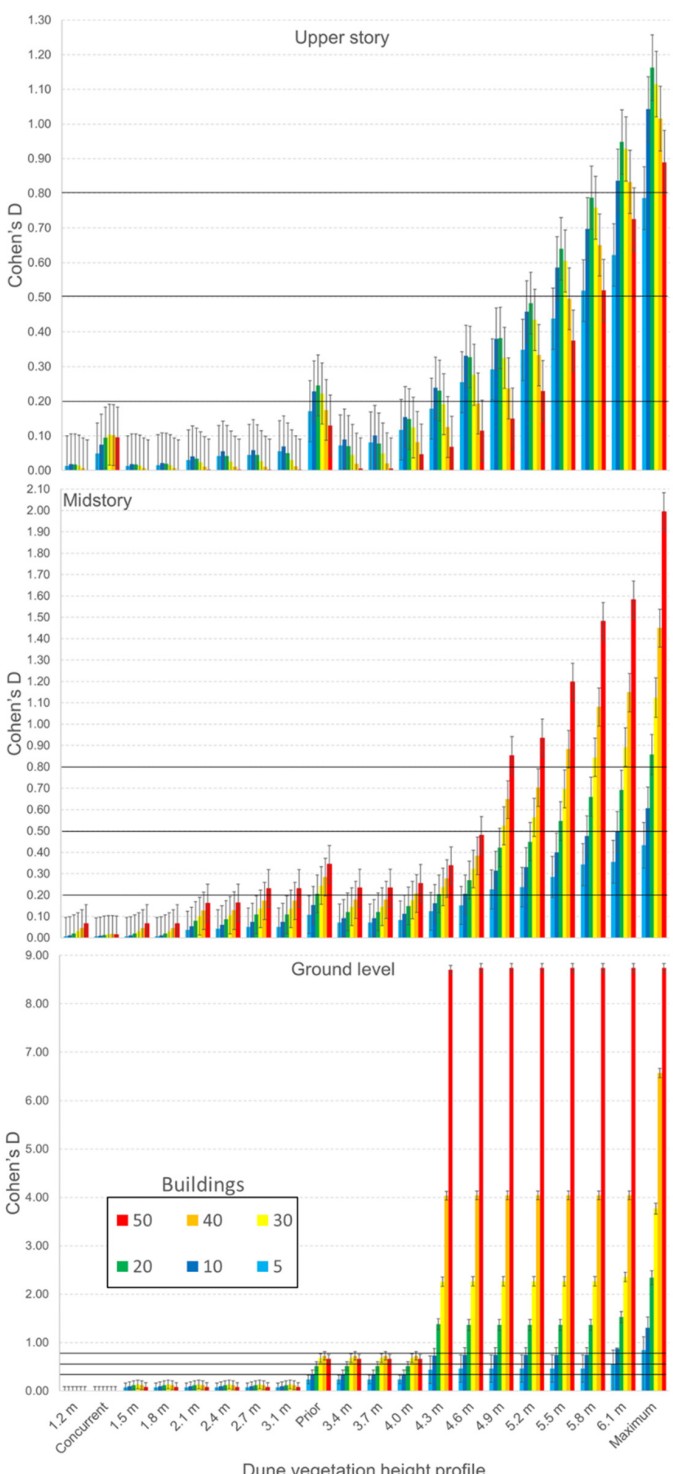

**Figure 4.** Mean Cohen's D effect sizes and their 95% confidence intervals for differences in numbers of turtle nests exposed to buildings based on viewshed analyses using seagrape height profiles derived from LiDAR data collected in 2018 (and 2004 for the prior-to-trimming vegetation profile) in coastal dunes in the study site in Delray Beach, Florida, USA. Differences in the numbers of nests exposed were determined by contrasting viewshed results for a trimmed vegetation height class (~1 m) to each taller vegetation height class in the graph. Viewsheds were run at three building heights (upper story, midstory, and ground level) for random subsamples of buildings (5–50 buildings; iterated 1000 times for each subsample) within the profiles of dune seagrape height. Dark horizontal lines indicate small (0.20–0.50), medium (0.51–0.80), and large (≥0.80) effect sizes.

### 3.4. Nonrandomly Selected Viewsheds

For the 10 buildings with the greatest individual differences in light visibility across vegetation profiles, light exposure from their cumulative model decreased in area (and in percent of beach area) from 18,726 m$^2$ (17.0%) for the trimmed profile to 18,381 m$^2$ (16.7%) for concurrent, 8499 m$^2$ (7.7%) for prior, and 3855 m$^2$ (3.5%) for maximum (Figure 5). For the 10 buildings with the smallest individual differences in light visibility across vegetation profiles, light exposure from their cumulative model for the trimmed profile was 20,212 m$^2$ (18.3%), compared to 20,206 (18.3%) for concurrent, 18,865 (17.1%) for prior, and 15,854 (13.1%) for maximum.

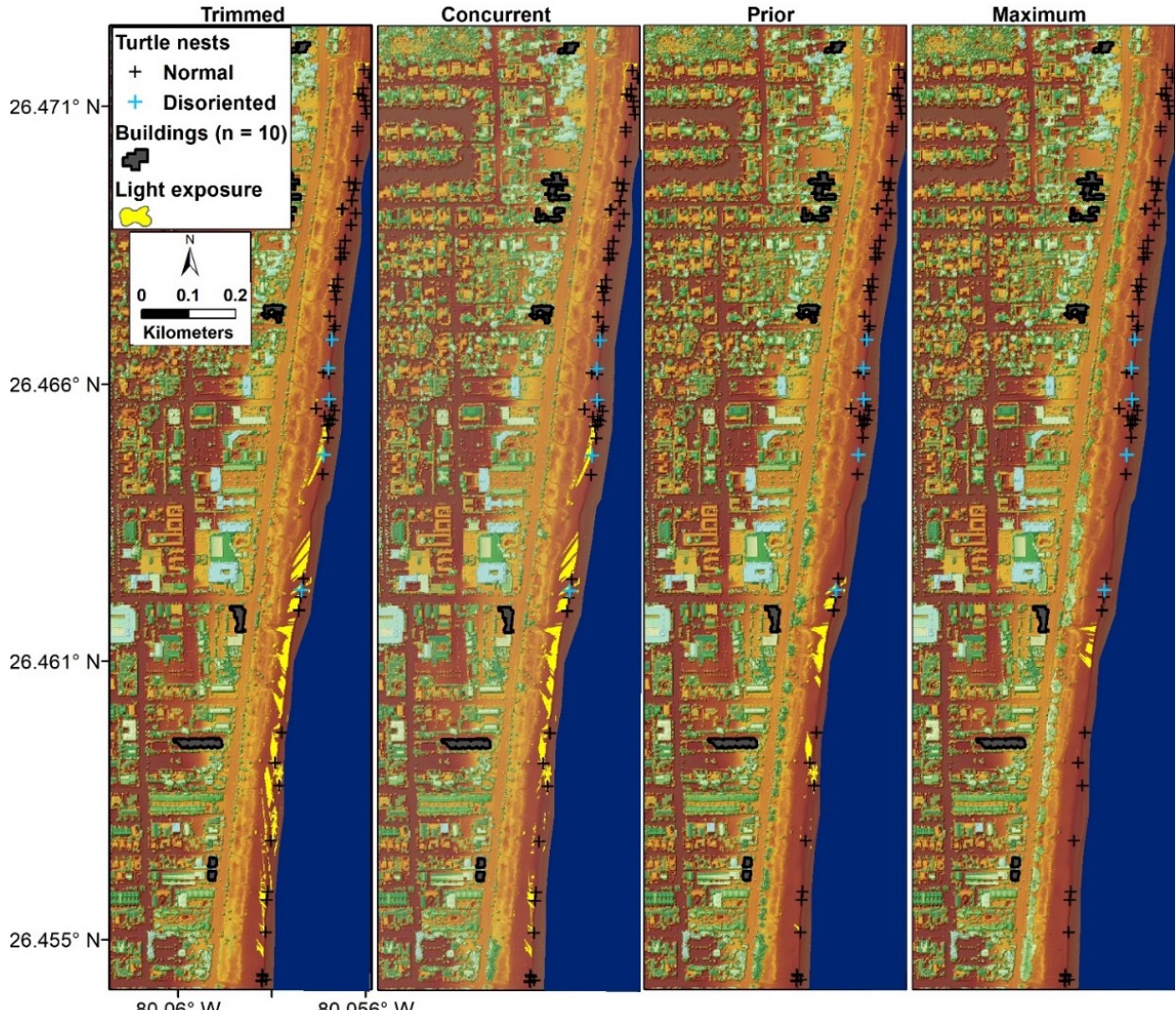

**Figure 5.** Examples of cumulative viewshed models depicting lighting visible on the beach from 10 nonrandomly selected buildings with all heights (upper story, midstory, ground level) combined, compared across four height profiles of dune vegetation in Delray Beach, Florida, USA, in 2018. Turtle nest data were collected in 2018 and depict five nests where hatchling disorientation was observed; however, the disorientation events are shown for reference only and do not necessarily associate with the example viewshed images. Dune vegetation profiles were created from LiDAR data (1 m resolution) and include trimmed to ~1 m tall, concurrent with marine turtle nest data in 2018, prior based on 2004 LiDAR data, and a generated maximum height between 6 and 8 m.

## 4. Discussion

Our viewshed models demonstrated that trimming beachfront vegetation can increase the exposure of marine turtle nesting areas to artificial lighting. Vegetation blocked some building lights in all untrimmed height profiles, especially for midstory and ground-level lighting. Taller vegetation (>4.6 m) was most effective in our randomly iterated models, but it is unlikely that all seagrape patches would reach their maximum height. For the prior-to-trimming vegetation profile, nonrandomly selected building lights were noticeably blocked across the beach (Figure 5), but the percentage of nests affected (generally <6%) and mean effect sizes (Cohen's D < 0.50) from randomly iterated models were relatively small (Figures 3 and 4). Lower effect sizes resulted from random iterations, including lights from 19 buildings (mainly taller ones) being consistently visible from the beach regardless of vegetation height. One tall building in our study area was visible to 33% of the beach area and could have exposed 81% of turtle nests to artificial light. The percentage of nests exposed can be considerably amplified if more buildings have lights on (e.g., 5 vs. 50 buildings in our models), especially under trimmed vegetation profiles. These findings further support the importance of proper vegetation management and the implementation of lighting modifications and regulations near marine turtle nesting beaches.

Impacts of artificial lighting on loggerhead turtles (e.g., decreased nest density) have been documented in other areas of Florida [10]. Loggerhead turtles tend to nest near vegetation and in areas where artificial light levels are moderate [11]. Much of the beach in our study area, however, had a high potential for lighting exposure limiting nest site selection in unlit areas (Figure 2). Although rates of hatchling disorientation before and after vegetation trimming were lacking at our study site, trimmed vegetation on the 2018 landscape allowed more light trespass on the beach, which might have interfered with hatchling movements. Our study area had an 8.9% rate of disorientation, which was greater than the mean rate of 2.7% for the state of Florida in 2018 [46] but slightly less than the recovery plan's goal of 10% [47]. On a nearby beach in Boca Raton, Florida, the orientation of turtle hatchlings was not disrupted by lights at lower heights or when the lighting was blocked by vegetation [22].

Dune vegetation can provide other benefits to marine turtles besides limiting disorientation to hatchlings. Sex determination occurs in the eggs of marine turtles and is temperature dependent, with increasing numbers of female hatchlings as temperatures increase, which is more marked with climate change [48–50]. The shading of nests by beach vegetation such as seagrapes may produce sex ratios that are less skewed toward females [51–55]. Intact dune vegetation also stabilizes dunes under the threat of erosion, wave run-up, or storm surge [56]. The Delray Beach vegetation-trimming plan [30] did include mitigative planting of shorter native plant species to help stabilize the dune [26,57]. Coastal vegetation can build and enlarge dunes over time, which could also shield inland lights and facilitate hatchling orientation toward the water [24,58].

Vegetation trimming will likely continue as a method of enhancing or maintaining property values and coastal views on Florida beaches; therefore, more strategic timing and manner of vegetation trimming should be considered. For example, for the prior-to-trimming profile, the mean vegetation height was smaller than those of other profiles, but the unevenness in height was greater, as seen in the standard deviation, which was almost doubled comparatively (Table 1). This led to a depression in the trend line in Figure 3, as taller seagrape patches in specific locations blocked more light coming from buildings. Preserving relatively tall and large patches of vegetation in strategic areas (instead of producing a uniformly trimmed height) could simultaneously meet both human and wildlife needs. However, strategic trimming could prove contentious if it led to improved ocean views for some properties while neighboring views remained impeded. Rotating which vegetation patches are trimmed over time could mitigate this issue, though it could be logistically challenging. Another option is to determine the peak nesting season and only trim vegetation afterwards, thus allowing vegetation to grow between large nesting events. However, if extensive vegetation trimming is proposed, then all lights

that are made more visible should be reviewed and modified as needed (e.g., shielded or changed to a long-wavelength bulb) to minimize light trespass, which can disorient hatchling turtles [14,59].

Viewshed analyses and remotely sensed data on light intensity have been used to measure the impacts of artificial lighting on marine turtle nesting beaches in other areas [7,9,10]. To our knowledge, however, our study is the first to incorporate vegetation trimming profiles with viewshed models. Our study is a novel approach for assessing the impacts of artificial lighting on marine turtles, but the models have some limitations. Our study did not investigate all sources of direct artificial lighting (e.g., streetlights) or all lighting factors (e.g., intensity, attenuation) or lighting durations (e.g., intermittent residential lighting vs. constant commercial or public safety lighting). Nor did we consider reflected light (e.g., skyglow), which also can disorient marine turtles [8]. Our models represented only potential lighting scenarios from buildings because we lacked lighting observations paralleling the 2018 nesting season. Our iterative viewshed models incorporated lights from entire buildings rather than individual lights per building (i.e., each light on a building could produce a different viewshed). A more realistic yet meticulous approach would be to run models on a per light basis, but this would call for 1206 viewsheds per model run (rather than 73), which would considerably increase the processing time and require substantial data-storage capabilities but likely would not provide much more useful information regarding general lighting impacts. Despite these limitations, our viewshed models provided new insights on potential light exposure that may occur from vegetation trimming.

## 5. Conclusions

Using a GIS, remotely sensed data, and viewshed modeling, our study examined the ramifications of the trimming of dune vegetation as it affects the amount of artificial light from urban buildings on a marine turtle nesting beach. Untrimmed vegetation can block more direct-line-of-sight lighting from buildings. However, regardless of vegetation height, the nesting beach would be exposed to artificial light emitted from tall buildings. If the option to erect shorter buildings is available to urban planners, it should be considered. Although we analyzed potential lighting impacts, our models can be adjusted to assess actual lighting impacts on marine turtles and other wildlife and can be applied to other nesting areas. Furthermore, our modeling approach could be used to evaluate changes in beach viewsheds before and after natural disasters (e.g., hurricanes) and other disturbances that might impact dune vegetation. Methods used in our study could be applied to research projects, lighting-proposal designs, urban planning (e.g., building height and placement), and vegetation-management plans to help reduce the impacts of light pollution on wildlife.

**Author Contributions:** Conceptualization, M.A.B. and K.N.S.; Methodology, M.A.B.; Software, M.A.B.; Validation, M.A.B. and K.N.S.; Formal Analysis, M.A.B.; Investigation, M.A.B. and K.N.S.; Resources, M.A.B. and K.N.S.; Data Curation, M.A.B. and K.N.S.; Writing—Original Draft Preparation, M.A.B. and K.N.S.; Writing—Review and Editing, M.A.B. and K.N.S.; Visualization, M.A.B.; Supervision, K.N.S.; Project Administration, K.N.S.; Funding Acquisition, K.N.S. All authors have read and agreed to the published version of the manuscript.

**Funding:** This research received no external funding.

**Data Availability Statement:** LiDAR data analyzed in our study are publicly available at https://coast.noaa.gov/dataviewer/#/ (accessed on 5 May 2021). Other data can be provided upon reasonable request to the corresponding author.

**Acknowledgments:** We thank the Florida Fish and Wildlife Conservation Commission's Imperiled Species Management (ISM) section, which provided the initial concept for this project to address its management needs, and which oversaw the field surveys and provided the data. We thank the marine turtle permit holders for collecting nesting data and Ecological Associates Inc., Jenson Beach, FL, USA), for collecting lighting data on behalf of ISM monitoring programs. We thank B. Bankovich, R. Baumstark, B. Crowder, T.M. Long, and R.N. Trindell for reviewing earlier versions of the manuscript.

**Conflicts of Interest:** The authors declare no conflict of interest.

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
