# Peer review of "Modeling Artificial Light Exposure after Vegetation Trimming at a Marine Turtle Nesting Beach"

_remotesensing, doi:10.3390/rs14112702_

Round 1

Reviewer 1 Report

The results of this research are important and have practical importance represented in clarifying The repercussions of pruning dune plants as it affects the amount of artificial light of urban buildings on the sea turtle nesting beach. Uncut plants can block more direct line-of-sight illumination from buildings. It provides useful scientific suggestion

Author Response

Although we did not see any specific comments from Reviewer 1 that needed to be addressed, we thank them for their review of our manuscript.

Reviewer 2 Report

Artificial light pollution is well-discussed problem nowadays. The author's aim to study modifications to the height of beach dune vegetation affect the amount of urban artificial lighting at a marine turtle nesting beach in Southeast Florida sounds interesting and promising. The paper is well planned, presented with high quality and sounds scientific. However, I would suggest to add more comparison of obtained results with scientists who studies the same question. Do the authors know such studies? Or if there were no such investigations, and authors do this at first time, they should write so. Also, some typography improvements are needed (authors could do this by their own): some legends for figures are on another page, some free space left on pages etc. All this remarks are minor and the paper are sustainable for being published.

Author Response

We are not aware of other peer-reviewed studies that specifically addressed impacts of vegetation trimming on artificial light exposure; however, we did state (as recommended by the reviewer) that our study was a novel approach in the Discussion (please see original manuscript: Page 15 lines 443-445).

 We removed extra spacing throughout the paper and aligned Figures with legends to be on the same page, though this might change at the final copyedit stage.

Thank you for your review.

Reviewer 3 Report

Dear authors,

please, find attach my review in the pdf file.

Kind regards

Author Response

Thank you for the detailed review of our manuscript.  We appreciate your comments and edits and believe they improved our manuscript. Please see the attached pdf of our responses.

Thank you.

Reviewer 4 Report

As stated in Results, this is an interesting and novel approach for assessing the impacts of artificial lighting on marine turtles, but the  the study did not investigate all sources of direct artificial lighting. Therefore the suggestion is to continue with this kind of study and to investigate all sources of direct artificial lighting in the next step of research.

Author Response

Although we did not see any specific comments that needed to be addressed, we thank you for your review of our manuscript.  We agree the study should be continued in the future to address other sources and factors of artificial lighting impacts on wildlife.

Round 2

Reviewer 2 Report

Accept in present form